# Digital data-based strategies: A novel form of better understanding COVID-19 pandemic and international scientific collaboration

Yan Wang[1]*, Henan Zhao[2]*

1 Scientific Research Center, The Second Hospital of Dalian Medical University, Dalian, Liaoning, China,
2 Department of Pathophysiology, College of Basic Medical Sciences, Dalian Medical University, Dalian, Liaoning, China

* zhaohenan@dmu.edu.cn (HZ); yanzi941026@126.com (YW)

## Abstract

International scientific collaborations have always been regarded as critical actions to address global pandemics, however, there was an obvious uncertainty between international collaboration and the COVID-19 control. We aim to combine digital data-based strategies to produce meaningful and advanced insights into the imbalance between COVID-19 and international collaboration, as well as reveal possible influencing factors, and ultimately enhance global collaboration. We conducted three retrospective cohort studies using respectively COVID-19 data from WHO, a complete dataset of scientific publications on coronavirus-related research from WoS, and daily data from Google Trends (GT). The results of geovisualization and spatiotemporal analysis revealed that the global COVID19 pandemic still remains serious. The global issue of imbalance between international collaborations and pandemic does exit, and the nations with good pandemic control had their own characteristics in above-mentioned correlation. Digital epidemiology provides, at least in part, evidence-based assessment and scientific advice to understand the imbalance between international collaborations and COVID-19. Our investigation demonstrates that transdisciplinary conversation through digital data-based strategies can help us fully understand the complex factors influencing the effectiveness of international scientific collaboration, thus facilitating the global response to COVID-19.

## Introduction

Coronavirus Disease 2019 (COVID-19) [1–3] with an exponential growth rate has become a global pandemic and hugely challenged the current healthcare systems of the world beyond expectations. At the time of writing (September 10, 2020), the cumulative number of globally confirmed cases is 27,766,325, and the globally cumulative deaths has reached 902,468 (World Health Organization, WHO). Unfortunately, so far, there are still no proven specific therapies available for COVID-19. Simultaneously, scientists self-organizing into collaborations, including international collaborations have been performed in determining its etiology, identifying

longitude values are collected from the WHO (https://www.who.int/) and Johns Hopkins University (https://coronavirus.jhu.edu/) as described in the paper.

**Funding:** Henan Zhao: This work was supported by Liaoning Education Department Program (LZ2019031) and Liaoning Provincial Program for Top Discipline of Basic Medical Sciences.

**Competing interests:** NO authors have competing interests.

the virus, defining the vulnerabilities that may allow treatment, and conducting research on drug and vaccine development [4].

Scientific collaborations have always been regarded as critical actions to address global pandemics [5, 6], since collaborations and/or international collaborations can not only make more nations coming into the global network to handle the crisis, but allow different nation or scientist to mutually access researching, working expertise or resources. While much can be gleaned from our general understanding of the importance of global collaboration and coordination in responding to COVID-19, many pertinent questions exist. As a previous report indicated that, due to political and economic considerations, the willingness to implement pandemic control measures varies from nation to nation [7]. In this regard, we have noticed the imbalance between international scientific collaboration and COVID-19 pandemic control, which means that increasing epidemiological research or global collaboration does not always lead to good epidemic control. This inevitably leads us to underline the corresponding shift in the perception of global collaboration and its implications. We pose that in the COVID-19 pandemic, there are indeed complex factors that influence the effectiveness of international collaboration, infection control and surveillance [8, 9].

Information and truths about the world and the contingent threats of infectious disease are increasingly extracted in the forms of digital signals and signs, rather than being generated from statistical processes only through human analysis [10]. Therefore, in this article, we took full advantage of digital data-based strategies, such as big data analysis and visualization, algorithmic techniques, scientometrics and digital epidemiology to produce meaningful and advanced insights into imbalance between COVID-19 and international collaboration, as well as revealed possible influencing factors. We hope our work can provide scientists, health workers, and even governments more effective new ideas for sharing experiences, accessing complementary expertise outside their country's borders, designing treatment strategies, effective vaccines and public-health policies, and ultimately effective speeding up the control of the pandemic spread.

## Methods

### Data source

Basic data sets of COVID-19 epidemiology, including officially confirmed, deaths, recovered, active, incidence-rate, case-fatality-ratio, daily reported, latitude, and longitude values, are all from the WHO (https://www.who.int/) and Johns Hopkins University (https://coronavirus.jhu.edu/). COVID-19 data was collected from January 22, 2020 to August 12, 2020.

In addition, in order to achieve the goals of the study, we first adopted the approach of using a complete dataset of scientific publications on coronavirus-related research between January 1, 2019 and August 22, 2020 from the Web of Science (WoS) as the basis for constructing the scientometric networks of disciplines, then analyzed and visualized the transdisciplinary collaboration networks in coronavirus research [4]. Any overlap between articles found across the different source materials was removed by CiteSpace (v5.7.R1). The following keywords were used in searches in the Title/Abstract/Keywords of each article in the respective databases: "COVID-19" OR "2019-nCoV" OR " SARS-CoV-2". Actually, the virus first appeared in the scholarly literature on January 24, 2020 [11].

In this work, we took advantage of Google Trends (GT) as the search query database (https://trends.google.com/trends/) for digital epidemic analysis, and used search keyword "lockdown" to analyze the effects of international cooperation on pandemic control, even including unreasonable effects in some nations. We obtained daily data from January 26, 2020 to August 9, 2020, which is within the time frame of the epidemic data.

## Data analysis & visualization

Referring to the previous researches [4, 12–14], the global collaboration networks were constructed based on international co-authorships. First, collaboration links were established by article addresses. Analysis matrices ("country/region" and "organizer") were all then created to show which countries or institutions are co-authoring articles together, respectively. Finally, in order to assess any changes in collaboration network, the degree, weighted degree, betweenness centrality and eigenvector centrality were calculated through software VOSviewer (v1.6.15) and Pajek (v5.09). CiteSpace and VOSviewer were used for visualizing the networks in the form of scientific landscapes.

Data visualization of COVID-19 epidemiology was performed using R software (v4.0.2) and Origin software (v2020b). Principal component analysis (PCA) and UMAP (Uniform Manifold Approximation and Projection) are performed using the FactoMineR and umap package of the R software. Matrix analysis and heatmap were performed using the corrplot package of the R software. Circos plot was performed using Perl software (v5.28.1) and Adobe Illustrator.

Geovisualization and its synthesis with epidemic modelling techniques were implemented by R software and Origin software. The world map was downloaded from the public domain of U.S. Geological Survey (USGS) National Map website (http://viewer.nationalmap.gov/viewer/).

## Statistical analysis

In Fig 3, we used 20 correlation coefficients as the indicators to perform statistical analysis on two cohorts composed of all 22 nations (Fig 3A left panel) and only nations with good pandemic control (the second cohort: Italy, China, Germany, France, Spain, Sweden, Belgium and Canada, Fig 3A right panel) through a two-tailed paired $t$-test. In addition, the correlation in matrix analysis was mainly carried out by using the corrplot package to perform Pearson correlation analysis through the software R. The correlation coefficient and significance ($p$-value) were extracted through programming statements cor and res. Statistical significance was established a priori as $p < 0.05$.

## Results

### Analysis of the current worldwide scenario and trends of COVID-19 through reconstruction of pandemic reprogramming in pseudospace and pseudotime manners

In order to understand the relationship between global collaboration and the COVID-19 pandemic, we first utilized big data strategies to analyze the current worldwide scenario and the trends of this novel pandemic.

Different from MERS- and SARS-CoV, COVID-19 comes under the subgenus *Sarbecovirus* of the *Orthocoronavirinae* subfamily and is a group 2B coronavirus [1–3]. As of 24:00 August 12, 2020 (UTC+8), a total of 20,620,847 confirmed cases of COVID-19 (with 749,358 deaths, and 7,393,767 active cases) have been reported in 186 affected countries and regions worldwide (WHO) (Fig 1). COVID-19 has been reported on all continents except Antarctica. As the two nations that experienced the earliest outbreaks of the virus, China and Italy have managed to control the COVID-19 pandemic in their respective nations. Simultaneously, the United States (US), India and Brazil have been the three major focus of concerns in terms of the large number of confirmed cases, deaths and active cases (Fig 1B and 1C upper panel). Johns Hopkins' interactive dashboard features real-time data has provided daily updates on the emerging novel coronavirus (COVID-19, https://coronavirus.jhu.edu/map.html).

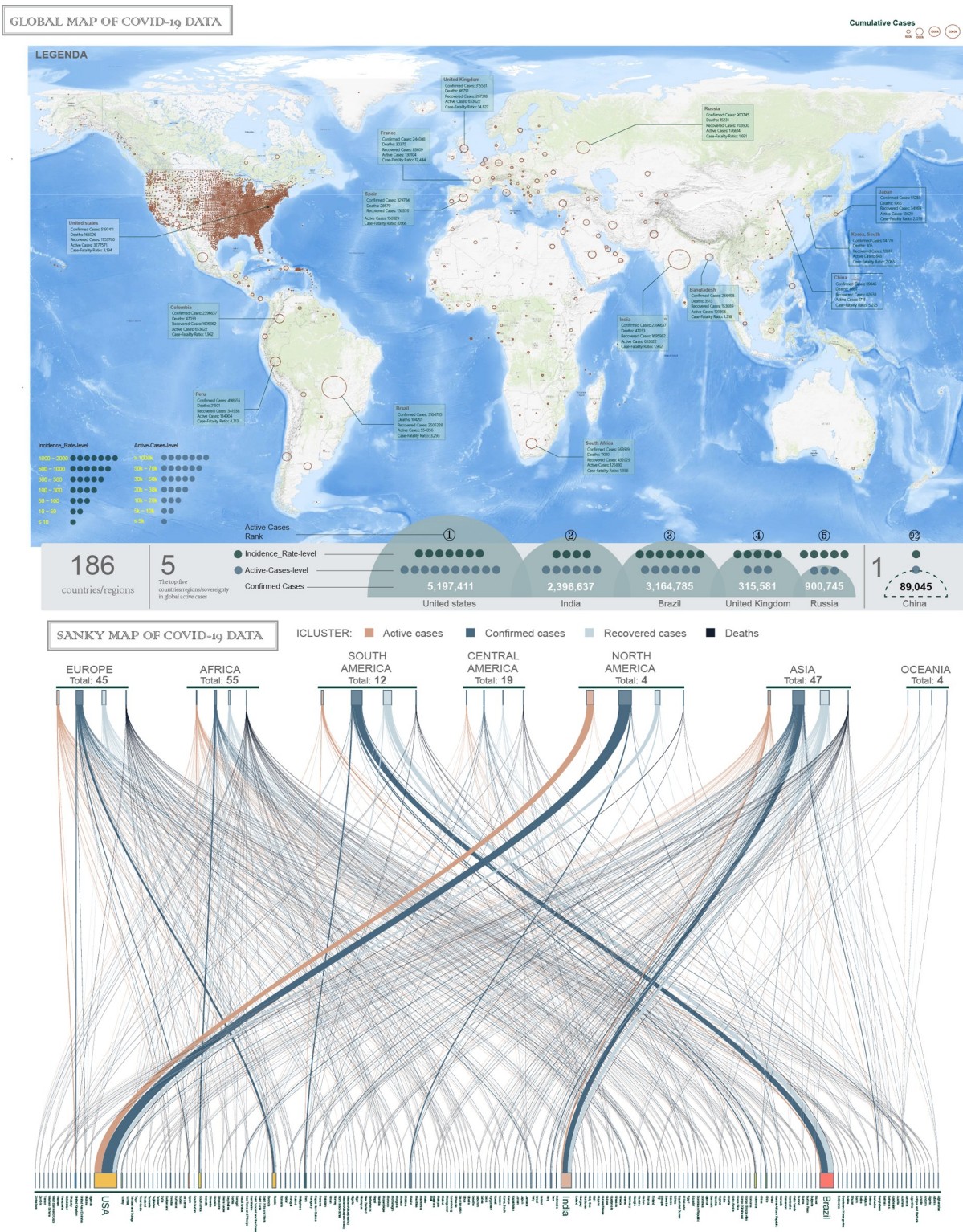

**Fig 1. World map depicting the current scenario of COVID-19 and analysis through reconstruction of pandemic reprogramming in pseudospace and pseudotime manners.** (A) Shown are 186 countries or regions with reported confirmed cases of COVID-19 as of August 12, 2020. Different colors indicate different WHO designated geographical regions with the number of active cases, and the diameter of the brown circle reflects the number of confirmed cases. The number of droplets just indicate the range of incidence-rate or active cases. (B) The WHO region-wise total number of confirmed cases, deaths, recovered cases and active cases are depicted in different color Sanky strips. (Based on data

from the WHO at https://covid19.who.int/table; updated numbers of cases, deaths, and patients recovered can be found at https://coronavirus.jhu.edu/map.html). (C) Reconstruction of pandemic reprogramming in pseudospace and pseudotime manners. The PCA results (for total 186 nations, upper panel) clearly show the confirmed cases, deaths, case fatality rates and geographic characteristics of each country. The UMAP analysis (for 22 selected nations, lower panel) provides strong warning trends for certain countries (shown in large font).

To avoid overlapping and unrecognizable analysis results due to the excessive number of samples and simultaneously based on international cooperation participation and epidemiological situations (confirmed, deaths and recovered), 22 nations were selected for UMAP analysis. UMAP, an algorithm constructed from a theoretical framework based in Riemannian geometry and algebraic topology, is wildly used in machine learning, and can arguably preserve more of the global structure with superior run time performance. Analyzing data were obtained from January 22, 2020 to August 12, 2020. We found that in addition to the United States, Brazil and India, the UMAP analysis also provided strong warning trends of COVID-19 for certain nations in pseudo-space, such as Colombia, Russia, Mexico, the United Kingdom (UK), South Africa, Iran, Pakistan and Japan (Fig 1C lower panel).

## International networks of scientific collaboration during the current COVID-19 period

Next, we analyzed the international networks of scientific collaboration during the current COVID-19 period. As expected, as of the time of writing, most nations, including China and Italy, have participated in coronavirus-related research and the international collaboration networks (Fig 2A upper panel and Table 1). Meanwhile, we also performed data visualization and PCA analysis for above 22 nations (Fig 2B and 2C). As seen in this visualization, the United States, China, Italy, the United Kingdom, Germany, France, and Canada have strong research linkages between nations, that is, research cooperation. Table 1 shows network metrics for major actors in the global network during the current COVID-19 pandemic, and the United States is the core player in the international network of collaboration. In terms of eigenvector centrality, the United States, the United Kingdom, Italy, China, and Germany are the top 5 nations in the network. Moreover, we found that the United States and China maintain the strongest connections in the collaborative network. We next turned the analysis from the national to the institutional level (Fig 2A middle panel and 2D-2F) to assess which institutions were the largest producers of coronavirus research in the COVID-19 period. Among the top 20 most prolific institutions, there were 7 US institutions, 5 Chinese institutions, and 3 UK institutions. Huazhong University of Science and Technology (which includes Tongji Hospital, Tongji Medical College, and Wuhan Union Hospital), located in Wuhan, China are the most prolific institutions during COVID-19, followed by the Harvard Medical School, and Wuhan University (Fig 2D). Through the unsupervised hierarchical clustering analysis of all institutions in selected 22 nations, it was found that China and the United States exhibited remarkably more similar collaborative performance in the international network during the crisis period (Fig 2E). Next, we benchmarked the performance of all institutions in the selected 22 nations. One of the most important of these findings is that, even with active pandemics, institutions in developing countries rarely participated in coronavirus-related research. (Fig 2F). Co-authorship network of authors shows that Chinese authors account for an obvious proportion of collaborations (Fig 2A lower panel), which is not a surprising, because during COVID-19 period, Chinese agencies were acknowledged as funding the majority of published papers [4].

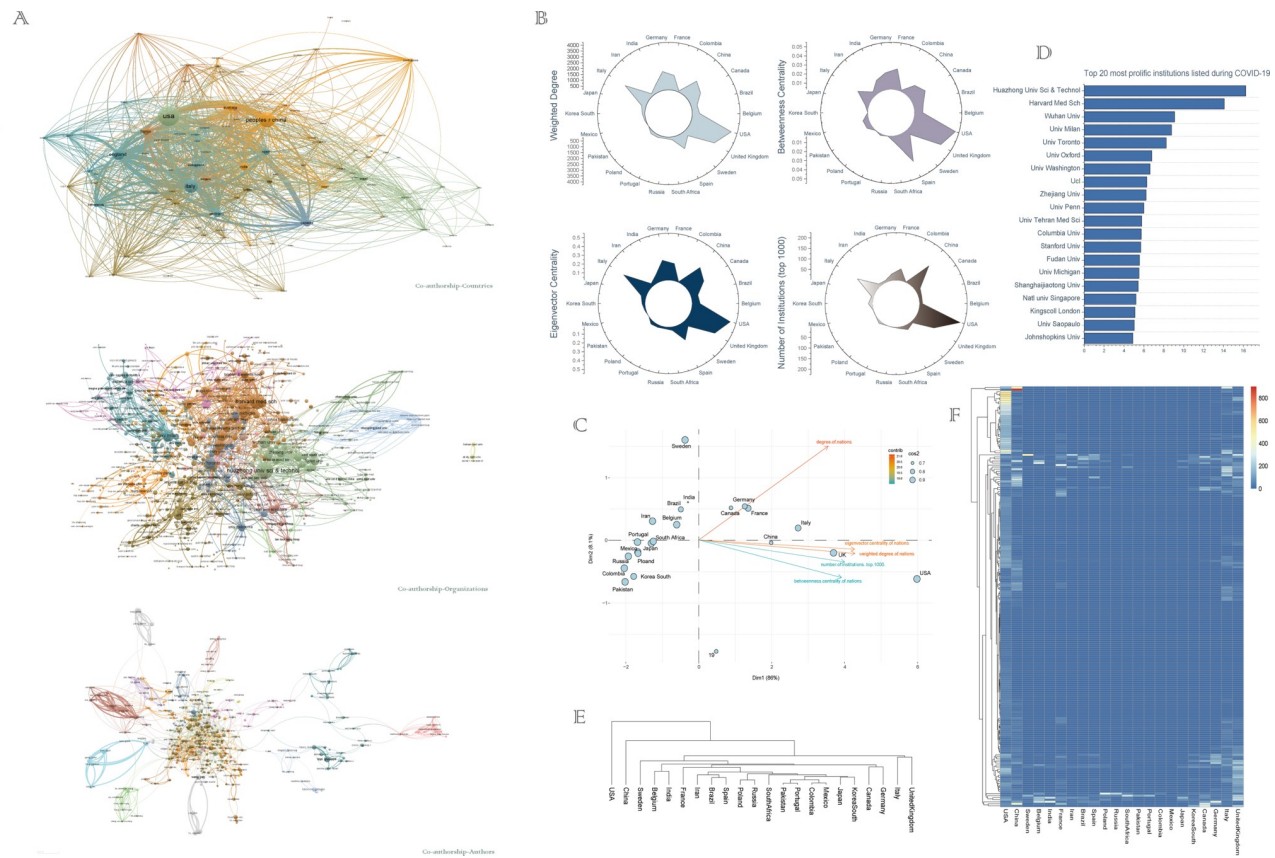

**Fig 2. Big data analysis and data visualization reveal the current situation of international scientific collaboration during the COVID-19 period.**
(A) International collaboration map of the COVID-19 research during the COVID-19 period. Upper panel: co-authorship network of countries; middle panel: co-authorship network of institutes; lower panel: co-authorship network of locations. (B-C) Circos map and PCA analysis of international collaboration networks in 22 nations during the COVID-19 period. (D) The top 20 most prolific institutions listed during COVID-19. (E-F) Unsupervised hierarchical clustering and heatmap displaying of the performance of all institutions in selected 22 nations.

## Imbalance between international collaboration and COVID-19 control

During the COVID-19 pandemic, current issues affecting the effectiveness of global collaboration are often depicted as essentially incalculable. Fortunately, matrix analyses provide very important information in this regard. Compared with all 22 nations (the first cohort, Fig 3A left panel), after separating, the characterizes of the correlations between bibliometrics parameters and pandemic parameters in nations with relatively better pandemic control (the second cohort: Italy, China, Germany, France, Spain, Sweden, Belgium and Canada, Fig 3A right panel) were different and had stronger interpretability than that in the previous cohort. In the second cohort, we found that international collaboration was conducive to the diagnosis of COVID-19, as well as reduced the number of active cases and increased the number of recovered cases by reducing the incidence-rate and case-fatality-ratio, respectively. The correlation coefficient was extracted through programming statements cor. Contrastingly, for the first cohort, the logics seemed chaotic. The circos chart further described the imbalance and uncertainty of international collaboration between these both cohorts from the perspective of data visualization (Fig 3B). Finally, we used 20 correlation coefficients as the indicators to perform statistical analysis on two cohorts composed of all 22 nations and only nations with good pandemic control, and the results showed that there was a statistical difference between the two cohorts (Fig 3C, two-tailed paired $t$-test, $p = 0.0405$, t = 2.199).

**Table 1. Network metrics of scientific collaboration for selected nations during COVID-19 period.**

| Country | Degree | Weighted degree | Betweenness Centrality | Eigenvector Centrality | Number of Institutions (top 1000) |
|---|---|---|---|---|---|
| Belgium | 78 | 815 | 0.012134 | 0.109537 | 15 |
| Brazil | 83 | 709 | 0.014322 | 0.098094 | 16 |
| Canada | 91 | 1413 | 0.021985 | 0.225935 | 25 |
| China | 88 | 1829 | 0.01711 | 0.29462 | 127 |
| Colombia | 59 | 278 | 0.004527 | 0.025905 | 2 |
| France | 95 | 1456 | 0.026083 | 0.210239 | 56 |
| Germany | 93 | 1794 | 0.02176 | 0.248339 | 32 |
| India | 86 | 845 | 0.01641 | 0.106435 | 10 |
| Iran | 75 | 396 | 0.007688 | 0.054064 | 13 |
| Italy | 95 | 2578 | 0.020927 | 0.35605 | 106 |
| Japan | 69 | 565 | 0.007007 | 0.076742 | 10 |
| South Korea | 58 | 418 | 0.003141 | 0.053622 | 20 |
| Mexico | 66 | 287 | 0.009615 | 0.03251 | 1 |
| Pakistan | 57 | 184 | 0.008845 | 0.018584 | 2 |
| Poland | 64 | 449 | 0.003286 | 0.057606 | 12 |
| Portugal | 67 | 406 | 0.004883 | 0.047871 | 2 |
| Russia | 62 | 320 | 0.003331 | 0.040377 | 2 |
| South Africa | 71 | 441 | 0.011341 | 0.059108 | 8 |
| Spain | 58 | 1474 | 0.036345 | 0.205377 | 24 |
| Sweden | 99 | 673 | 0.009016 | 0.089902 | 9 |
| United Kingdom | 99 | 2820 | 0.042358 | 0.371218 | 102 |
| USA | 107 | 3761 | 0.051135 | 0.47917 | 206 |

## Digital epidemiology can be regarded as a new approach of understanding the uncertainty in "global collaboration-pandemic control"

We compared the pandemic curve of dates from January 26, 2020 to August 9, 2020, which can cover GT data and WHO epidemiological data, to analyze the matrix relationship between Internet keyword search volume and the number of confirmed infections and active cases. Nine nations were selected, representing the European and Asian countries that are currently being affected or have been severely affected by the pandemic. Visual inspection of the correlation coefficient between the GT search volume and the corresponding epidemiological data showed certain rules. (Fig 4A). The correlation coefficient and significance (*p*-value) were extracted through software R. Statistical significance was established as $^*p < 0.05$, $^{**}p < 0.01$ and $^{***}p < 0.001$.

In order to gain insights into the significance of above correlation coefficients, hierarchical clustering analysis and PCA analysis were further performed. Since the search keyword "lockdown" can reflect the public and/or the government's concerns to a certain extent, the results of hierarchical clustering analysis indicated that the United States, the United Kingdom, Brazil, India and Japan were relatively similar or close in terms of related concerns, while Italy, Germany, China and South Korea were more similar or close (Fig 4B). PCA analysis further interpreted the above-mentioned differences and the influencing factors of these differences from the 2-D pseudo space (Fig 4C).

These findings indicate that the analysis results of digital epidemiology have strong similarities with epidemic situation and risk prediction, suggesting that in addition to global collaboration, there are other factors that are also important indicators for epidemiological prevention and control. Identifying and revealing them and then taking targeted measures can help us

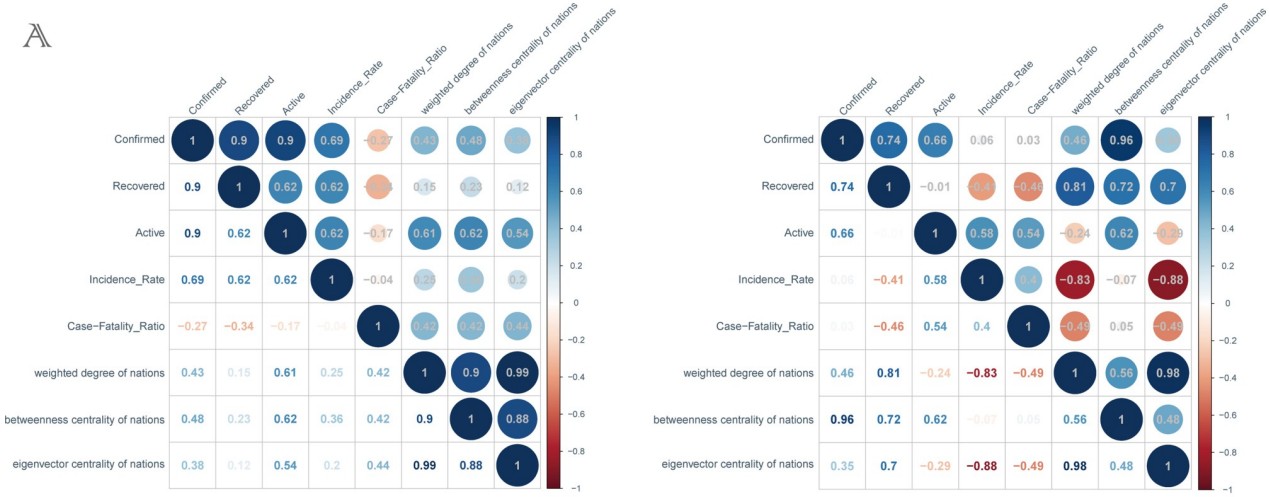

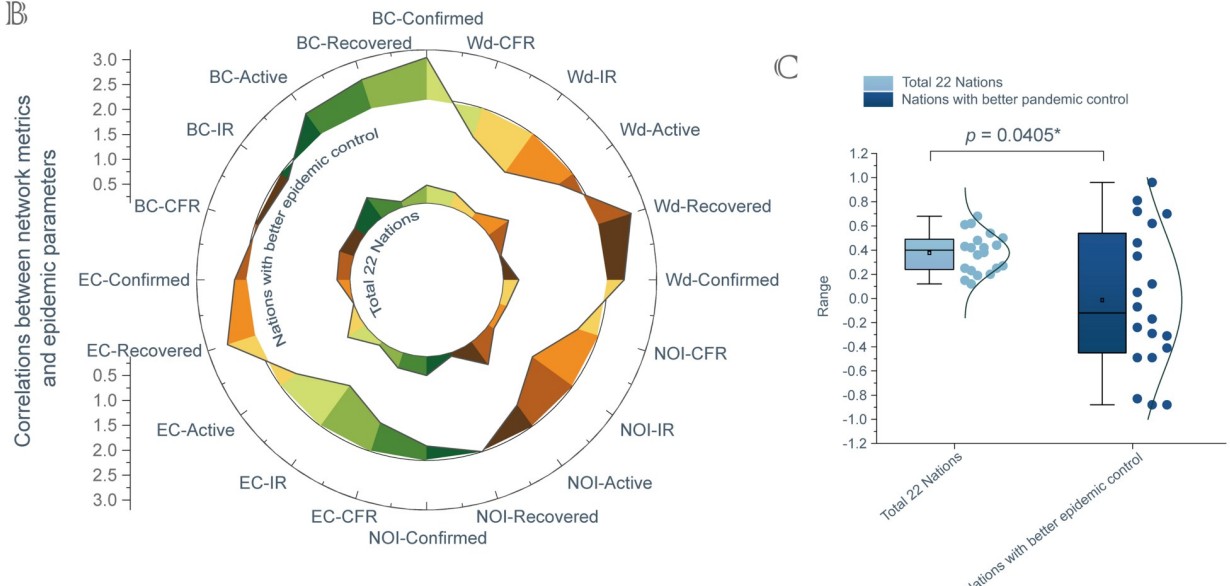

**Fig 3. The imbalance between international collaboration and COVID-19 control.** (A) Matrix analysis of the correlation between network metrics and pandemic parameters. left panel: 22 selected nations, right panel: nations only with good pandemic control. (B) Circos plot to show the correlation between network metrics and pandemic parameters for both cohorts, respectively. (C) Results of two-tailed paired $t$-test are $^*p < 0.05$, t = 2.199, n = 20. Wd: weighted degree, BC: betweenness centrality, EC: eigenvector centrality, NOI: number of institutions (top 1000), IR: incidence-rate, CFR: case-fatality-ratio; Wd-Confirmed: the correlation between weighted degree and confirmed cases, and so on.

improve the global collaboration more effectively. A strategy based on digital data is a good way to achieve this goal, at least in part.

## Discussion

Recently, a novel coronavirus, SARS-CoV-2, causing COVID-19, emerged in late 2019, has posed a global health threat, causing an ongoing pandemic in many nations and regions. Simultaneously, health workers worldwide are currently making efforts to control further outbreaks caused by COVID-19 virus.

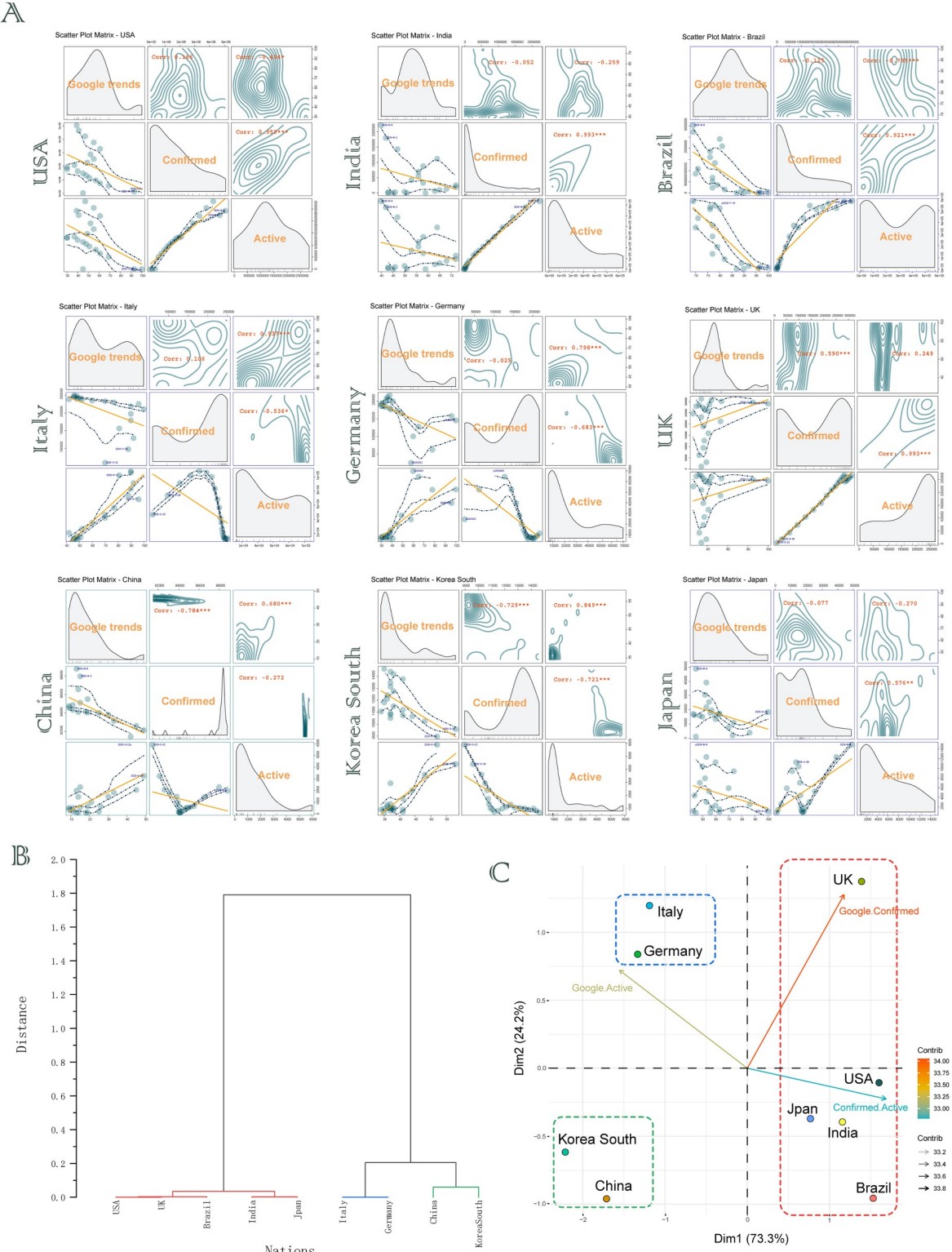

**Fig 4. Digital epidemiology is a new way to understand "global collaboration-pandemic control" uncertainty.** (A) Matrix analysis of correlation of GT (search keyword "lockdown") and WHO epidemiologic data (the number of confirmed cases and active cases respectively). (B) Hierarchical clustering analysis of above correlation. (C) PCA analysis of above correlation. The color of the dashed box represents different hierarchical clusters. *Corr*: correlation coefficient, *p<0.05, **p<0.01, ***p<0.001.

Since the onset of the global pandemic, the United States, China, Italy, France, Germany and the United Kingdom, *etc.*, as well as institutions/organizations, researchers and health workers in their respective nations have actively participated in international networks of collaboration. Especially, the United States and China maintained the strongest connections in this collaborative network. Based on this logic, in our research, we hypothesize that a global pandemic, especially the pandemic in related nations, will lead to a reduction. However, the analyses based on digital-data strategies revealed a more complex picture, suggesting that there were indeed complex factors affecting international cooperation and the effectiveness of infection control and surveillance.

Additionally, the ascendancy of the era of big data and the rise of digital disease have afforded unprecedented new opportunities towards the enhancement and supporting of disease analysis and prediction capabilities that are increasingly focused on international scientific collaboration. In turn, these new digital data-based strategies will lead to new methods and tools, and interdisciplinary in the use of those methods and tools will run at full capacity to connect the operational dots between those seemingly unintelligible and largely unstructured streams of data in the non-digital/physical world, thereby allow for a new interpretation of the complex dynamics of international collaborations. In this study, the combination of these new digital data-based strategies (i.e., big data analysis and visualization, algorithmic techniques as well as digital epidemiology), has produced meaningful and advanced insights into the scientific tracking and predictive analysis of COVID-19 pandemic, the patterns of international collaboration in COVID-19 research, as well as the understandings of the form, character and effectiveness of international collaboration in this crisis.

It is worth mentioning that over the past decade, digital epidemiology has become an integral part of public health surveillance to monitor communicable diseases [15], in order to better understand diseases and public concerns, perceptions, and behaviors on health issues [16]. Digital epidemiology can be broadly defined as epidemiology that uses digital methods from data collection to data analysis [15, 17, 18]. In 2009, researchers from Google and the US Centers for Disease Control and Prevention (CDC) published a method to estimate flu activity by region using search queries [15]. Up to date, GT has been regarded and served as a prime data source for digital epidemiology. Digital epidemiology superimposes another layer onto this, and to some extent, also comprises information sourced from policy, guidance on public or public health from government.

Importantly, dissecting the underlying factors and mechanisms of COIVD-19 is the fundamental for informing the public-health policies needed to limit disease spread. In other words, scientific research is the basis of all measures, while international collaboration can help scientists, health workers, and even governments share experiences, access complementary expertise outside their country's borders, design treatment strategies, effective vaccines and public-health policies, and then speed up the control of the pandemic spread.

Admittedly, the potential and reliability of openly available online content remain largely unknown and constantly changing, as well as digital data-based strategies only provide preliminary evidence of an emerging problem, but they hold potential for collective wisdom, highlight possibilities, amplify positive signals supported by evidence and science, and help the scientists, institutions or organizations generate hypotheses to guide and prioritize future research on potential mechanisms, provide scientific advice, as well as policymakers of government formulate the public-health policies.

Finally, we use a graphical abstract to summarize the current relationship between international cooperation and COVID-19 in this article (Fig 5), so as to provide a reference for us to better understand the relationship between them. In summary, novel techniques to the

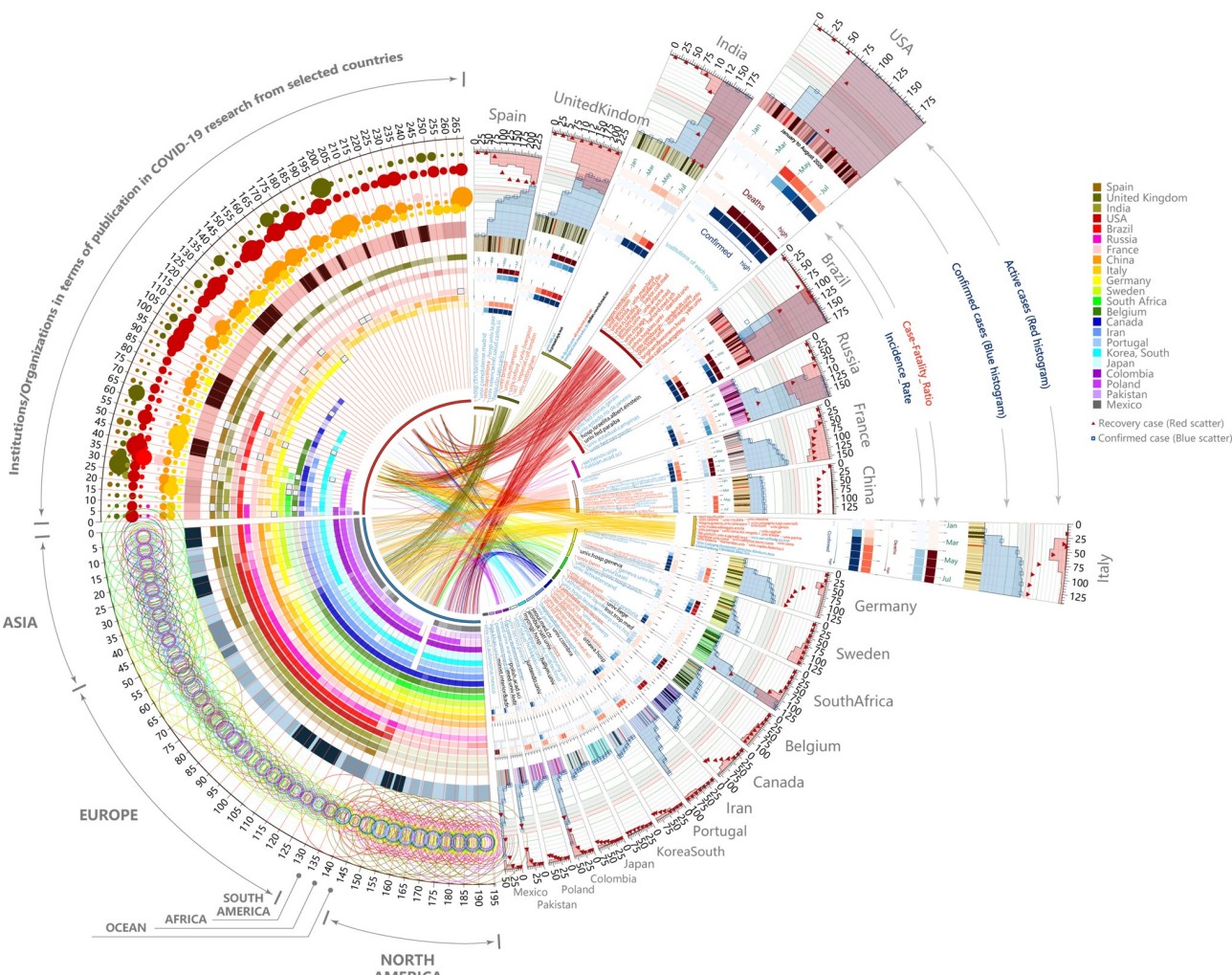

**Fig 5. Quick glance over the current status of international collaboration and the COVID-19 pandemic, and the "global collaboration-pandemic control" imbalance.** Scientific collaborations have always been regarded as critical actions to address global pandemics, however, we have noticed that in some nations, the increasing number of epidemiological studies or global collaboration has not brought well pandemic control, while in others the opposite has been the case. Based on the international cooperation participation and epidemiological situations, 22 nations were selected. The upper-left panel shows weight distribution of institutions/organizations in 22 nations; The lower-left panel shows the weight distribution of institutions/ organizations in the 6 continents; the right-half panel shows the outbreaks and important institutions/organizations in 22 nations. There seems to be no clear logics between the international cooperation participation and epidemiological situations.

pandemic threats via these digital data-based strategies seem to emerge epistemological shifts, as well as methodological changes and multiple social and political influences [10]. The objective of this work is to highlight the potential gains and benefits yielded by new data sources and digital processing techniques, and finally help all nations respond to the COVID-19 or the preempt looming pandemic.

## Conclusion

Transdisciplinary conversation can help us fully understand the complex factors that affect the effectiveness of international scientific collaborations, thereby facilitate the global response to COVID-19, and digital data-based strategies are a good way to achieve this goal.

## Acknowledgments

We thank Dr Jianmei Ma, the members of the Ma lab for critical feedback and discussion. We also thank Shimin Yang (the first affiliated hospital of Dalian Medical University, grade 2017) for part of the data verification work.

## Author Contributions

**Conceptualization:** Henan Zhao.

**Data curation:** Yan Wang, Henan Zhao.

**Formal analysis:** Yan Wang, Henan Zhao.

**Funding acquisition:** Henan Zhao.

**Investigation:** Yan Wang, Henan Zhao.

**Methodology:** Yan Wang, Henan Zhao.

**Software:** Yan Wang, Henan Zhao.

**Visualization:** Henan Zhao.

**Writing – original draft:** Yan Wang, Henan Zhao.

**Writing – review & editing:** Henan Zhao.

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
