## [Decision Letter · Decision Letter 0]

23 Feb 2021

PONE-D-20-37110

Digital data-based strategies: a novel form of better understanding COVID-19 pandemic and international scientific collaboration

PLOS ONE

Dear Dr. Zhao,

Thank you for submitting your manuscript to PLOS ONE. After careful consideration, we feel that it has merit but does not fully meet PLOS ONE’s publication criteria as it currently stands. Therefore, we invite you to submit a revised version of the manuscript that addresses the points raised during the review process.

We look forward to receiving your revised manuscript.

Kind regards,

Pasquale Avino, Ph.D.

Academic Editor

PLOS ONE

Journal Requirements:

3. We note that Figure 1 in your submission contains map images which may be copyrighted.

We require you to either (a) present written permission from the copyright holder to publish this figure specifically under the CC BY 4.0 license, or (b) remove the figure from your submission:

b. If you are unable to obtain permission from the original copyright holder to publish this figure under the CC BY 4.0 license or if the copyright holder’s requirements are incompatible with the CC BY 4.0 license, please either i) remove the figure or ii) supply a replacement figure that complies with the CC BY 4.0 license. Please check copyright information on all replacement figures and update the figure caption with source information. If applicable, please specify in the figure caption text when a figure is similar but not identical to the original image and is therefore for illustrative purposes only.

4. Please include your tables as part of your main manuscript and remove the individual files. Please note that supplementary tables should be uploaded as separate "supporting information" files.

Additional Editor Comments:

Dear Authors, please follow the referees' suggestions and accurately revise the language throughout the manuscript. With my best regards, Prof. Pasquale Avino

Reviewers' comments:

Reviewer #1: 1. Add references in Introduction part for lines 21-29.

2. Statistical analysis should be explained in more detail.

3. In the reference section, all references should be in similar format.

4. English should be improved for this manuscript.

---

## [Author Response · Author response to Decision Letter 0]

5 Mar 2021

Reviewers' comments:

Reviewer #1:

1. Add references in Introduction part for lines 21-29.

Dear professor, thank you for your work to help us improve our manuscript. Because in the first submission, we forgot to add the line number into our paper, in the revised version, we add new references to the two paragraphs with lines between 21-29 to improve the credibility and readability of our paper. 

We add the new reference #8 and #9 to illustrate that the factors affecting the pandemic are diverse and very complex. 

8. Wilder-Smith A, Freedman DO. Isolation, quarantine, social distancing and community containment: pivotal role for old-style public health measures in the novel coronavirus (2019-nCoV) outbreak. J Travel Med. 2020;27(2). Epub 2020/02/14. doi: 10.1093/jtm/taaa020.

9. Bowers KW. Balancing individual and communal needs: plague and public health in early modern Seville. Bull Hist Med. 2007;81(2):335-58. Epub 2007/09/12. doi: 10.1353/bhm.2007.0020. 

We add the reference #10 (its original reference number was #16) to illustrate that information and truths about the world are increasingly being extracted in the forms of digital signals and signs, rather than being generated from statistical processes. 

2. Statistical analysis should be explained in more detail.

We first add more detail of statistical analysis into the "Statistical analysis" of the "Methods" section, including the statistical objects and statistical analysis packages involved in the programming software. Next, in Figs 3 and 4 in the "Results" section, we also added more detail into the statistical analysis, including the statistical methods, p-value and t-value. In the correlation analysis of the matrix, the programming package and code of correlation coefficient and p-value were introduced at the same time. 

3. In the reference section, all references should be in similar format.

We have organized the format of the references according to your suggestions and PLOS ONE's style requirements. Thank you.

4. English should be improved for this manuscript.

We have carefully checked and revised our paper sentence by sentence, including language usage, spelling, and grammar. Thank you again.

---

## [Editor Report · Decision Letter 1]

16 Mar 2021

Digital data-based strategies: a novel form of better understanding COVID-19 pandemic and international scientific collaboration

PONE-D-20-37110R1

Dear Dr. Zhao,

We’re pleased to inform you that your manuscript has been judged scientifically suitable for publication and will be formally accepted for publication once it meets all outstanding technical requirements.

Kind regards,

Pasquale Avino, Ph.D.

Academic Editor

PLOS ONE

---

## [Editor Report · Acceptance letter]

22 Mar 2021

PONE-D-20-37110R1 

Digital data-based strategies: a novel form of better understanding COVID-19 pandemic and international scientific collaboration 

Dear Dr. Zhao:

I'm pleased to inform you that your manuscript has been deemed suitable for publication in PLOS ONE. Congratulations! Your manuscript is now with our production department. 

Kind regards, 

on behalf of

Professor Pasquale Avino 

Academic Editor

PLOS ONE